

# Intercomparison of Photoacoustic and Cavity Attenuated Phase Shift Instruments: Laboratory Calibration and Field Measurements

Jialuo Zhang[1], Jun Chen[1]*, Meng Wang[1], Mingxu Su[1], Wu Zhou[1], Ravi Varma[2], Shengrong Lou[1,3]*

[1]Shanghai Key Laboratory of Multiphase Flow and Heat Transfer in Power Engineering, School of Energy and Power Engineering, University of Shanghai for Science and Technology, Shanghai 200093, China

[2]Department of Physics, National Institute of Technology Calicut, Calicut 673601, Kerala, India

[3]State Environmental Protection Key Laboratory of the Cause and Prevention of Urban Air Pollution Complex, Shanghai Academy of Environmental Science, Shanghai 200070, China

*Correspondence to*: Jun Chen (j.chen@usst.edu.cn)

Shengrong Lou (lousr@saes.sh.cn)

**Abstract:** The study of aerosol optical properties is essential to understand its impact on the global climate. In our recent field measurement, a photoacoustic extinctiometer (PAX) and a cavity attenuated phase shift albedo monitor (CAPS-ALB) were used for online aerosol optical properties measurement. Laboratory calibration with gas and particle samples were carried out to correct disagreements of field measurements. During particle calibration, we adopted ammonium sulfate (AS) samples for scattering calibration of nephelometer parts of both the instruments, then combined with number-size distribution measurements into MIE model for calculating the value of the total scattering (extinction) coefficient. During gas calibration, we employed high concentration $NO_2$ for absorption calibration of PAX resonator, then further intercompared the extinction coefficient of CAPS-ALB with a cavity-enhanced spectrometer. The correction coefficient obtained from the laboratory calibration experiments was employed on the optical properties observed in the filed measurements correspondingly, and showed good result in comparison with reconstructed extinction from the IMPROVE model. The intercomparison of the calibrated optical properties of PAX and CAPS-ALB in field measurement were in good agreement with slopes of 1.052, 1.024 and 1.046 for extinction, scattering and absorption respectively, which shows the reliability of measurement results and verifies the correlation between the photoacoustic and the cavity attenuated phase shift instruments.

**Key words:** Aerosol optical properties, instrument calibration, photoacoustic spectroscopy, cavity



attenuated phase shift spectroscopy

## 1 Introduction

Atmospheric aerosols can directly affect the earth's energy balance and cause global temperature
changes by absorbing and scattering solar radiation(Horvath, 1993;Haywood and Shine, 1995;Penner et
al., 2001). Therefore, considerable studies were undertaken to investigate the optical properties of aerosol
particles from different regions(Baynard et al., 2007;Petzold et al., 2013;Moosmüller et al., 1998). The
optical properties of regional aerosols depend on particle size distribution, mixing state and complex
refractive index, thus online measurements are necessary(Nakayama et al., 2015;Schwartz et al., 2010).
Furthermore, the calibration of instruments is a key step to ensure the reliability and quality of online
measurement data of aerosol optical properties.
Ideally, the complete set of aerosol optical properties are required measuring simultaneously,
including aerosol extinction, scattering and absorption coefficients, for aerosol optical closure studies.
The integrating nephelometry (IN) is an effective, economical and widely recognized method for online
obtaining aerosol scattering coefficient(Beuttell and Brewer, 1949;Heintzenberg and Charlson,
1996;Abu-Rahmah et al., 2006). Early on the systematic limitations of this technique were noted, that is
so-called truncation error caused by technically impossible to cover the full range of the scattering angle,
and which has mainly studied through numerical simulations with Mie model(S. Ensor and P. Waggoner,
1970;Anderson et al., 1996;Anderson and Ogren, 1998;Heintzenberg et al., 2006;Müller et al., 2009).
The measurement techniques for the extinction coefficient of atmospheric aerosols mainly include cavity
ring-down spectroscopy (CRDS) technique, cavity attenuation phase shift (CAPS) technique and cavity
enhanced absorption spectroscopy (CEAS) technique. CRDS has extremely high detection accuracy and
mature    measurement    system,    which    performed    well    in    laboratory    studies    and    field
measurements(O'Keefe and Deacon, 1988;Baynard et al., 2007;Berden et al., 2010;Pettersson et al.,
2004;Strawa et al., 2003). Related in its basic principle to CRDS, previously CAPS was used to calibrate
the reflectivity of mirrors also applied to measure atmospheric nitrogen dioxide(Kebabian et al., 2005;Ge
et al., 2013;Herbelin and McKay, 1981). It currently has been extended to the field of aerosol extinction
coefficient measurement(Kebabian et al., 2007;Petzold et al., 2013). Massoli et al. (2010) gave a detailed
description of CAPS results in the aerosol extinction coefficient measurements, including the first





laboratory characterization and field deployment. Onasch et al. (2015) calibrated the optical path length
error of CAPS with MIE model using monodisperse polystyrene spheres generated in the laboratory.
Rather than single wavelength measurements, CEAS with broadband light source applied for
atmospheric trace gas detection(Fiedler et al., 2003;Ball et al., 2004;Chen and Venables, 2011) was later
extended to quantitative aerosol extinction(Varma et al., 2013;Zhao et al., 2014;Suhail et al., 2019). The
filter-based methods are most commonly used for online measuring aerosol absorption
coefficient(Horvath, 1997;Hansen et al., 1982;Petzold and Schönlinner, 2004). Considering aerosol
morphology changes, multiple scattering and shielding effects, these methods require many correction
factors that limits the quality of measurement results(Bond et al., 1999;Weingartner et al., 2003).
Recently, the photoacoustic spectroscopy (PAS) technique(Terhune and Anderson, 1977;Bruce and
Pinnick, 1977;Adams et al., 1990), a direct method that can be easily-calibrated, has been developed into
a stable instrument in the field measurement of aerosol absorption(Moosmüller et al., 1998;Arnott et al.,
1999;Lack et al., 2006;Lewis et al., 2008;Sharma et al., 2013;Nakayama et al., 2015). Arnott et al. (2000)
calibrated their aerosol photoacoustic instrument by measuring the photoacoustic response in the
presence of $NO_2$ and compared its result with aethalometer. Lack et al. (2006) used ozone with a known
optical absorption level to calibrate the photoacoustic system with CRDS.

75       During our recent field campaign in Yangtze River Delta (YRD), the measurements of aerosol

optical properties showed discrepancies from different instruments, among which the extinction,
absorption, and scattering coefficients were measured by CAPS, PAS, and IN respectively(Du et al.,
2020). For investigation of the discrepancy between instruments and correction of the measurement data,
this study carried out an aerosol optical properties intercomparison measurement. During calibration
measurement, the extinction coefficient was calibrated with MIE model using mono-disperse particles
and the absorption coefficient was calibrated with transmission method using an absorbing gas, while
the scattering coefficient was calibrated with combination of above model and the method using no-
absorbing particles. In addition, an Incoherent Broad-Band Cavity Enhanced Absorption Spectroscopy
(IBBCEAS) setup was used to measure extinction coefficient of $NO_2$ for comparing with CAPS. Then
the correction factors obtained from the laboratory calibration experiments were employed on the data
observed in the filed measurement correspondingly and compared with the reconstructed extinction of
the interagency monitoring of protected visual environment (IMPROVE) model. Furthermore, the
calibrated field measurement results from photoacoustic and cavity attenuated phase shift instruments
were intercompared. For aerosol optical properties, different optical methods showed good agreement
and closure correlation after calibration, which has been rarely studied in laboratory calibration and field
measurement. In addition, the corrected field measurement data are more reliable for subsequent study
of aerosol optical properties in YRD region.
**2 Materials and Methods**
**2.1 Instrument description**
During calibration experiments, the optical properties of aerosol were measured by a Cavity
Attenuation Phase Shift-ALBedo monitor (CAPS-ALB) (Shoreline Science Reaserch, Japan) and a
Photoacoustic Extinctiometer (PAX) (Droplet Measurement Technologies, US). In addition, a Scanning
Mobility Particle Sizer Spectrometer (SMPS) (Model 3938, TSI, US) was employed to measure the
number-size distribution for MIE model, and an IBBCEAS setup was used to measure $NO_2$ concentration
for extinction calculation. Above instrument details are summarized in Table 1.

**Table 1 Instrument Details**

| Instrument | Parameters | Time resolution | Flow, $Lmin^{-1}$ | Wavelength, nm |
|------------|------------|-----------------|-------------------|----------------|
| CAPS-ALB | Extinction coefficient, Scattering coefficient [$Mm^{-1}$] | 1 s | 0.85 | 530 |
| PAX | Absorption coefficient, Scattering coefficient [$Mm^{-1}$] | 1 s | 1 | 532 |
| SMPS | Number size distribution [$cm^{-3}$] | 5 min | 0.3 | - |
| IBBCEAS | $NO_2$ concentration [ppb] | 1 s | 0.6 | 355-380 |

Aerosol sample flow was drawn into the PAX using an external vacuum pump, then split between
the wide-angle integrating reciprocal nephelometer and photoacoustic resonator for simultaneous online
measurements of light scattering coefficient and absorption coefficient. In the photoacoustic cavity, the
laser beam passing through the sample stream was modulated at the resonant frequency of the cavity, and
the light-absorbing molecules were heated and quickly transferred the heat to the receiving end of the
instrument, the pressure wave generated by periodic heating wasdetected by a sensitive microphone. The


calculation formula of absorption coefficient ($b_{abs}{}^{obs}$) is as follows (Rosencwaig, 1980):
$$b_{abs}{}^{obs} = \frac{P_{mic} \cdot A_{res} \cdot \pi^2 \cdot f_{res}}{P_L \cdot (\gamma - 1) \cdot Q} \qquad (1)$$
Where, $P_{mic}$ is the pressure at the microphone at the resonant frequency $f_{res}$, $P_L$ is laser power, $A_{res}$ is the
geometric cross-section of the resonator, $\gamma$ is the ratio of specific heat at constant pressure and volume,
$Q$ is the quality factor of the resonator that calculated from temperature, pressure, and relative humidity
($RH$).
The wide-angle integrating reciprocal nephelometer with a scattering integration angle of 6-174°
range used in PAX, which detects scattering light from a parallel beam through a cosine-weighted
detector. The detector located in the center of the cavity is fiber coupled to a photo-multiplier tube (PMT),
where the measured laser power is proportional to the total scattering cross section. The expression for
determining scattering coefficient ($b_{sca}{}^{obs}$) is given by (Abu-Rahmah et al., 2006):
$$b_{sca}{}^{obs} = \frac{P_{PMT}}{P_L} \qquad (2)$$
Where $P_{PMT}$ is the value of the PMT signal with scattering background subtracted, $P_L$ is measured laser
power. The scattering background was measured during the zeroing process of the instrument operation.
In addition, the extinction coefficient ($b_{ext}{}^{obs}$) considered as theoretical value in recommended
calibration method of PAX that can be obtained by measuring the intensity of transmitted light with a
photodetector combined with Lambert Beer's law as follow:
$$b_{ext}{}^{obs} = \frac{ln \, (I_0/I)}{L} \cdot 10^6 [Mm^{-1}] \qquad (3)$$
Where, $I_0$ and $I$ are the laser intensity with or without extinction substances, respectively. $L$ is the path
length of the laser beam through the cavity in meters, here is 0.354 m. $10^6$ is a conversion factor to
express extinction in Mm$^{-1}$.
The CAPS-ALB using an internal vacuum pump to introduce aerosol flow into the sample cell to
measure the extinction coefficient and scattering coefficient simultaneously. Nearly 1° truncation angle
integrating sphere integrating nephelometer (ISIN) has been employed in CAPS-ALB. The integrating
sphere with attached truncation reduction tubes located around the sample cell and PMT are equipped to
collect scattering light, which effectively reduces the angle truncation error(Varma et al., 2003). As a
typical kind of reciprocal nephelometer, its scattering coefficient ($b_{sca}{}^{obs}$) can also be calculated using Eq

135 (2).

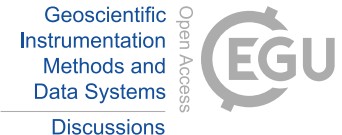

The extinction measurement system of CAPS-ALB utilizing a visible light-emitting diode (LED)
with the luminescence as a light source and a sample cell incorporating two high reflectivity mirrors
centered at the wavelength of the LED and a vacuum photodiode detector. The extinction coefficient of
CAPS-ALB ($b_{ext}^{obs}$) is obtained by measuring the light attenuation of the visible long optical path with a
vacuum photodiode, and detecting the phase shift of the square wave frequency modulation heterodyne
detection of the light source, its expression as follow:

$$b_{ext}^{obs} = (cot\,\vartheta - cot\,\vartheta_0) \cdot (2\pi f/c) \tag{4}$$

where $cot$ is the cotangent, $c$ is the speed of light, $f$ is the LED modulation frequency, $T$ and $P$ are the
sample temperature and pressure, respectively. The amount of phase shift ($\vartheta$) is a function of fixed
instrument properties such as cell length, mirror reflectivity and modulation frequency, and of the
presence of aerosols (Kebabian et al., 2007). The term $cot\vartheta_0$ is obtained from a periodic baseline
measurement (using particle-free air). It is worth mentioning that the effective optical-path error in the
sample cell of CAPS-ALB, which caused by the purge airflow of the mirror limits the space of the aerosol
samples, has been initially corrected in the internal calculation process. The original correction factor
was 0.7 that close to the value reported by Onasch et al. (2015), which generally calibrated with MIE
model calculation.
Our self-developed IBBCEAS device was used to measure gas concentration in the $NO_2$ comparison
experiment(Chen and Venables, 2011). The IBBCEAS measures the light intensity change of the light
source through the optical cavity, then inverts the concentration of the gaseous samples. When a pair of
high-reflectivity plano-concave mirrors with a reflectivity of $R$ are composed of an optical cavity with a
length of $L$ that is illuminated by continuous broadband incoherent light, the output light intensity $I$ is
equal to the sum of the output light intensity of each order. Combined with Lambert Beer's law, the
expression for extinction coefficient $b_{ext-CEAS}$ at measured wavelength as follow (Fiedler et al., 2003;Ball
et al., 2004):

$$b_{ext-CEAS}(\lambda) = \left(\frac{I_0(\lambda)}{I(\lambda)} - 1\right)\left(\frac{1 - R(\lambda)}{L}\right) = \Sigma\sigma_i(\lambda)N_i \tag{5}$$

Here, $I_0$ is light intensity without absorbing matter, $\sigma_i$ and $N_i$ are absolute extinction cross section
and concentration of species i. $I_0$, $I$, $R$, $\sigma_i$ and $N_i$ are functions of wavelength. Therefore, for different
detection wavelengths, the extinction coefficient cannot be compared directly. A simple method is to
establish a relationship with the species concentration. Fitting the extinction cross-section $\sigma_i$ to the



extinction coefficient ($b_{ext\text{-}CEAS}$), the concentration of the measured gas $N_i$ can be inverted. Noting that the
reflectivity $R$ of the cavity mirrors in IBBCEAS has been calibrated before our experiments, so the result
of IBBCEAS can be considered as absolute value.
The number size distribution for MIE model calculation was obtained from SMPS, which is consists
of an Electrostatic Classifier (Model3082, TSI, US) and a Condensation Particle Counter (CPC) (Model
3750, TSI, US). The Electrostatic Classifier was used with a Long Differential Mobility Analyzer (LONG
DMA) (Model 3081, TSI, US), its particle-size selection range is 14.1~736.5 nm, with a sample flow of
0.3 L min$^{-1}$ and a sheath flow of 3 L min$^{-1}$. The aerosol sample passes through the radioactive neutralizer
to be charged, then enters the DMA to selects particles of different particle sizes by changing the voltage.
The number of selected particles is counted to after the process of hygroscopic growth in CPC, which
has an uncertainty of within ±10 % in measuring particle concentration (Petzold et al., 2013).
**2.2 Experimental**
Based on the above principles, we adopted the following experimental procedures to compare PAX
and CAPS-ALB as Fig. 1 shows. The blue solid line represents the process of particle calibration, the red
solid line represents the procedures of gas calibration. All joints have been leak tested to ensure tightness.

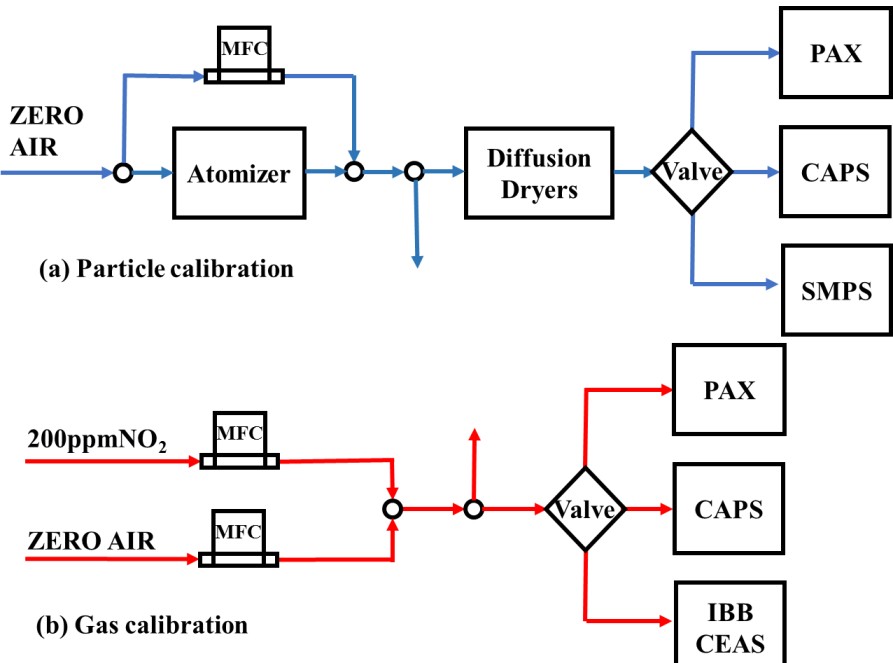


181            **Figure 1: Experimental schematics (a) Particle calibration (b) Gas calibration.**

a. Particle calibration
For systematic errors, such as angle truncation, laboratory generated nebulized ammonium sulfate
(AS) (AR 99%, Aladdin Chemical) aerosols were used to calibrate and test the nephelometers of CAPS-
ALB and PAX using same experimental procedure as follow. AS aqueous solution was nebulized by an
atomizer (Model 9032, TSI, US) with filter air at a constant inlet pressure of 20 psi, which can generate
a stable outlet flow rate of ~ 5 L min$^{-1}$. As shown by the blue solid line in the Fig. 1(a), the nebulized
aerosol flow was diluted with filtered air to adjust its concentration, and then dried using diffusion dryers
with silica gel that reduced the sample $RH$ to ~10 % before delivery to the instruments, where the excess
airflow was discharged by bypass. Only opening PAX or CAPS-ALB valves, the dry aerosol flow was
connected to the instruments sampling port for at least ~5 min until the measured value stabilizes. The
entire flow system used conductive silicone tubing's and reduced bending to minimize the loss of
particles during aerosol transportation. For high-concentration of non-absorbing AS aerosol with
refractive index of 1.53+0.00i, the absorption effect can be ignored. Therefore, the scattering calibration
factor ($f_{sca}{}^{obs}$) was calculated by comparing the measured extinction coefficient ($b_{ext}{}^{obs}$) and scattering
coefficient ($b_{sca}{}^{obs}$) (Lewis et al., 2008;Cross et al., 2010).
For the purpose of estimating the scattering or extinction coefficients measured in above
experiments and further correcting the absolute total scattering (extinction) coefficient, we performed an
additional calibration using polystyrene latex (PSL) spheres with Mie model. This model is a rigorous
analytical solution of the scattering field distribution of monochromatic light illuminates on spherical
particles (Born and Wolf, 1999). Thus, assuming the particles to be round, it is considered feasible to
apply Mie model to retrieve the number size distribution for calculating the total scattering coefficient of
atmosphere aerosol. The scattering and extinction efficiency factor $Q_{sca}$ and $Q_{ext}$ can be calculated from
the function of particle complex refractive index, light source wavelength and size distribution (Wu et
al., 2018;Bohren and Huffman, 1983). By integrating the particle cross-sectional area $\pi D^2/4$, particle
number concentration $N(D)$, and $Q_{sca/ext}$ on the particle diameter $D$ distribution, yields the calculated
scattering and extinction coefficient $b_{sca/ext}{}^{MIE}$ as follow expression:

$$b_{sca}{}^{MIE} = \int_0^\infty Q_{sca} \cdot \frac{\pi D^2}{4} \cdot N(D) \cdot dD \qquad (6)$$


$$b_{ext}{}^{MIE} = \int_0^\infty Q_{ext} \cdot \frac{\pi D^2}{4} \cdot N(D) \cdot dD \qquad (7)$$



The experiments incorporating mono-disperse PSL spheres with complex refractive index
1.60+0.00i and diameter of $350\pm6$ nm (Thermo Scientifc) were carried out follow the calibration
procedures of scattering calibration (replace PSL with AS). Opened the SMPS valve and connected the
diluted dry aerosol flow to the its sampling port, then continuously measured together with CAPS-ALB
or PAX for ~20 min for collecting at least three sets of effective data of particle-size distribution at each
concentration. By comparing the MIE calculated average with the measured value for multiple
concentrations, the MIE model correction factor ($f_{sca}^{PAX\text{-}MIE}$, $f_{ext}^{CAPS\text{-}MIE}$) can be determined.
b. Gas calibration
As noted in previous studies(Arnott et al., 2000), the PAS resonator acoustic calibration used
sufficiently high concentrations of absorbing gas to generate a huge absorption, so that the Rayleigh
scattering was negligible. Therefore, our experiment adopted high concentration $NO_2$ for absorption
calibration and determined the absorption correction factor ($f_{abs}^{obs}$) from comparison of measured
absorption ($b_{abs}^{obs}$) and extinction ($b_{ext}^{obs}$) coefficients without knowing $NO_2$ concentration. In the case
of only PAX valve opened, by diluting 200 ppm $NO_2$ in different dilution ratios, the filtered air and $NO_2$
mixture were introduced to PAX for ~5 min, in which the flow of filtered air and $NO_2$ were controlled
by the mass flow controller to specified proportion, respectively. The entire flow system used Teflon
tubes to minimize $NO_2$ loss and contaminations, and a bypass was set to ensure the stability of the sample
flow and pressure.
Subsequently, considering the possible particulate loss of CAPS-ALB calibration, IBBCEAS and
CAPS-ALB was used to measure $NO_2$ samples simultaneously for comparing the measured extinction
coefficient in gaseous way. This experiment was carried out based on the experimental procedure for
PAX absorption calibration, though closing PAX route and simultaneously opening the valves of CAPS
and IBBCEAS. Based on the limitation of IBBCEAS the $NO_2$ concentration was controlled below 1 ppm
and each concentration was maintained for at least about 15 min until the measured value stabilizes. By
multiplying the $NO_2$ concentration measured by IBBCEAS and the $NO_2$ extinction cross section from
previous study of Voigt et al. (2002) at the CAPS-ALB detection wavelength (530nm), the conversion
result of the extinction coefficient ($b_{ext\text{-}CEAS}$) measured by IBBCEAS was obtained. Thus, the extinction
correction factor ($f_{ext}^{CAPS\text{-}CEAS}$) from comparison with IBBCEAS can be determined.





**3. Results and Discussion**
**3.1 PAX calibration results**
In our calibration experiments for PAX, with assumption of linearity in calibration down to the
detection limit of the instruments (Arnott et al., 2000), the high concentration of absorbing gas and
scattering particles generated a huge absorption and scattering effect that weaken the interference of noise
for corrected the response curve of the PAX photoacoustic resonator and nephelometer respectively.
Fig. 2(a) shows the relationship between the absorption coefficient ($b_{abs}^{obs}$) and the extinction
coefficient ($b_{ext}^{obs}$) in the $NO_2$ measurement results. The slope of fitted line in Fig. 2(a), which represent
$f_{abs}^{obs}$, were determined to be $0.961 \pm 0.019$ with correlation factor $R^2 \sim 0.985$. The calibration result
showed that the absorption measurement of PAX only needs slight correction and has high accuracy. Fig.
2(b) presents typical correlation plots comparing the extinction coefficient from transmissivity ($b_{ext}^{obs}$)
for AS samples (The black solid dot) and MIE model calculation ($b_{ext}^{MIE}$) for PSL samples (The red solid
dot) with the measured scattering coefficient ($b_{sca}^{obs}$) respectively in PAX scattering calibration, where
the extinction and scattering are theoretically equivalent due to negligible absorption.

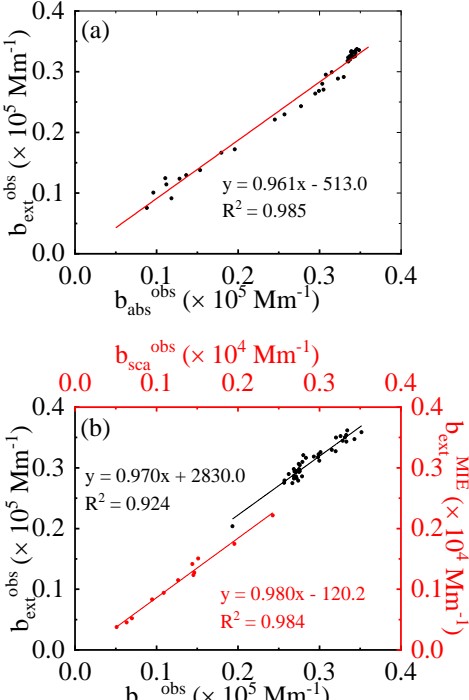


**Figure 2: PAX calibration results: (a) Comparison of the measured extinction and absorption**
**coefficient. (b) Comparison of the measured and MIE-model calculated extinction coefficient with the**
**measured scattering coefficient**
In Fig. 2(b), the slope of the black solid line indicates the measured scattering correction factor
($f_{sca}{}^{obs}$) that was determined to be 0.970±0.046 with correlation factor $R^2$~0.924. Moreover, we calculated
the absolute extinction coefficient with MIE model for further correction. Here, limited by the detection
range, another set of coordinate system was used for comparison. The slope of the red solid line that
indicates the MIE model scattering correction factor ($f_{sca}{}^{PAX-MIE}$) were determined to be 0.980±0.039 with
correlation factor $R^2$~0.984. The scattering correction factors from transmission method and MIE model
were within acceptable range of the truncation error, and had only ~1 % discrepancy in different
measurement range, showing that a good agreement between the two methods and the reliability of PAX
scattering calibration result.
**3.2 CAPS-ALB calibration results**
In the CAPS-ALB calibration experiment, we first utilized PSL spheres to correct its extinction
coefficient through MIE model calculation, and then employed AS samples to correct its scattering
coefficient comparing the calibrated extinction coefficient. In addition, we used self-developed
experimental IBBCEAS device for further verifying the correction factor calculated by MIE model.
Fig. 3(a) shows correlation of extinction measured by CAPS-ALB ($b_{ext}{}^{obs}$) and extinction calculated
by MIE model ($b_{ext}{}^{MIE}$) for 350nm mono-disperse PSL spheres. The slope in Fig. 3(a) represents
extinction correction factor ($f_{ext}{}^{CAPS-MIE}$) were determined to be 0.983±0.018 with correlation factor
$R^2$~0.999. It shows that the good accuracy of original calibration factor for the effective optical path error,
only slight adjustment was required. The other factors that might affect the extinction calibration is the
uncertainty of the aging effects of LED and detectors (PMT and vacuum photodiode), which has different
effects according to cell geometry.
Correlation plots comparing scattering coefficient ($b_{sca}{}^{obs}$) and extinction coefficient ($b_{ext}{}^{CAPS}$) for
AS samples measured by CAPS-ALB are shown in Fig. 3(b). According to its linear fitting result, $f_{sca}{}^{obs}$
were determined to be 1.016±0.002 with correlation factor $R^2$~0.996. It showed that the measured
scattering coefficient has high accuracy, and verified the reliability of extinction correction factor of
CAPS-ALB ($f_{sca}{}^{CAPS-MIE}$).
Fig. 3(c) presents the comparison between measured extinction coefficient of CAPS ($b_{ext}{}^{CAPS}$) and





IBBCEAS ($b_{ext}^{CEAS}$) for NO2 samples. The slope of Fig.3(c) represents extinction correction factor
($f_{ext}^{CAPS-CEAS}$) were determined to be 0.946±0.007 with correlation factor $R^2$~0.998. The experimental
correction factor of IBBCEAS ($f_{ext}^{CAPS-CEAS}$) was consistent with the theoretical correction factor of the
MIE model ($f_{ext}^{CAPS-MIE}$) within an acceptable error range of 4 %, proving that the reliability of MIE model
calculation and the applicability of CAPS-ALB calibration, no matter whether choosing gas or particle
ways.

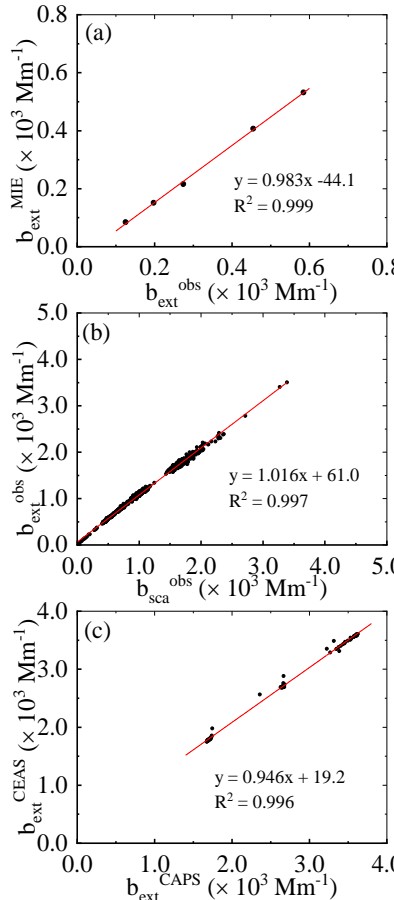


**Figure 3: CAPS-ALB calibration results: (a) Comparison of the measured and MIE-calculated**
**extinction coefficient. (b) Comparison of the measured extinction and scattering coefficient. (c)**
**Comparison between measured extinction coefficient of CAPS-ALB and IBBCEAS**
**3.3 Calibrated field measurement**

293        The field measurements were carried out in the Gehu area of southwest Changzhou City, Jiangsu


Province (31°63′ *N*, 119°90′ *E*) from 25 May to 27 June before the rainy season in 2019. Changzhou has
a location in the center of the Yangtze River Delta and has a subtropical monsoon climate. The
measurement site was surrounded by 60 % of ecological wetlands and green gardens, and 20 % of
territorial waters, which results represented the regional ambient conditions of the Yangtze River Delta
before the rainy season. The sampling point was located on the top floor of a building at the height of 15
m above ground and all sampling tubes used a cyclone size cutter (URG, 2.5 μm, 5 lpm).
The correction factor obtained from the laboratory calibration experiments was employed on the
optical properties observed in the filed measurement correspondingly. For comparison, the IMPROVE
model was applied to identify major chemical components contributing to light extinction during field
measurement (Pitchford et al., 2007;Tao et al., 2014), where major chemical components including
water-soluble inorganic ions, organic carbon (OC), and elemental carbon (EC) were analyzed and
quantified by a monitor for aerosols and gases in ambient air (MARGA) (ADI 2080, Metrohm,
Switzerland) and an OC/EC analyzer (Model RT-4, Sunset, US). The simplified general formula of
IMPROVE model used in reconstruction of total scattering (extinction) coefficient ($b_{ext}^{IMP}$) can be
expressed as(Xia et al., 2017):
$$b_{ext}^{IMP} = 2.2 \times f_s(RH) \times [Small\ (NH_4)_2SO_4] + 4.8 \times f_L(RH) \times [Large\ (NH_4)_2SO_4$$
$$+ 2.4 \times f(RH) \times [Small\ NH_4NO_3] + 5.1 \times f(RH) \times [Large\ NH_4NO_3]$$
$$+ 2.8 \times [Small\ OM] + 6.1 \times [Large\ OM] + 1.7 \times f_{ss} \times [SS] + 1.0 \times [FS]$$
$$+ 0.6 \times [CM] + 8.28 \times [EC]$$
$$[Large\ X] = [Total\ X]^2/20, [Total\ X] < 20,$$
$$[Large\ X] = [Total\ X], [Total\ X] \geq 20,$$
$$[Small\ X] = [Total\ X] - [Large\ X] \tag{8}$$
where [X] represent the mass concentration of aerosol chemical component X, μg/m$^3$; Ammonium
Sulfate [(NH$_4$)$_2$SO$_4$] = 1.375 [SO$_4^{2-}$]; Ammonium Nitrate [NH4NO3] = 1.29 [NO$_3^-$]; Organic Matters
[OM] = 1.6 [OC] ; Sea Salt [SS] = 1.8 [Cl$^-$]; Fine Soil [FS] = 2.2 [Al] + 2.49 [Si] + 1.94 [Ti] + 1.63 [Ca]
+ 2.42 [Fe]; Coarse Mass [CM] = [PM10] − [PM2.5]; $f_s(RH), f_L(RH)$ and $f_{ss}$ represent RH growth curves
of sulfate, nitrate, and SS (Jung et al., 2009). Due to the lack of soil element information, Ca$^{2+}$ was
assumed to account for 5 % of the concentration of fine soil mass based on previous studies, thus [FS] =
20[Ca$^{2+}$] (Amato and Hopke, 2012).



Considering unavailable period of aerosol composition measurement (Due to status of MARGA),
only from 1 to 6 June were selected for the comparison. Fig. 4 (a) and (b) showed intercomparison of the
measured extinction coefficient of PAX and CAPS-ALB with IMPROVE-calculated extinction
coefficient, the linear fitting slopes are 1.182 and 1.183 with the correlation factor $R^2$ of 0.807 and 0.824,
respectively. Comparing the correlation factor, it is in good agreement with Shanghai (0.83) and
Hangzhou (0.81) in previous studies (Wang et al., 2016). Thus, it can be concluded that the IMPROVE
model has good applicability in Gehu area. Here, the extinction of PAX was the sum of the measured
absorption and scattering. In addition, Fig. 4(c) showed a timing diagram of the extinction coefficient
from PAX, CAPS-ALB measurement and IMPROVE model calculation. It showed a good agreement
between the measured and theoretical value and proved the reliability of our measurement data.

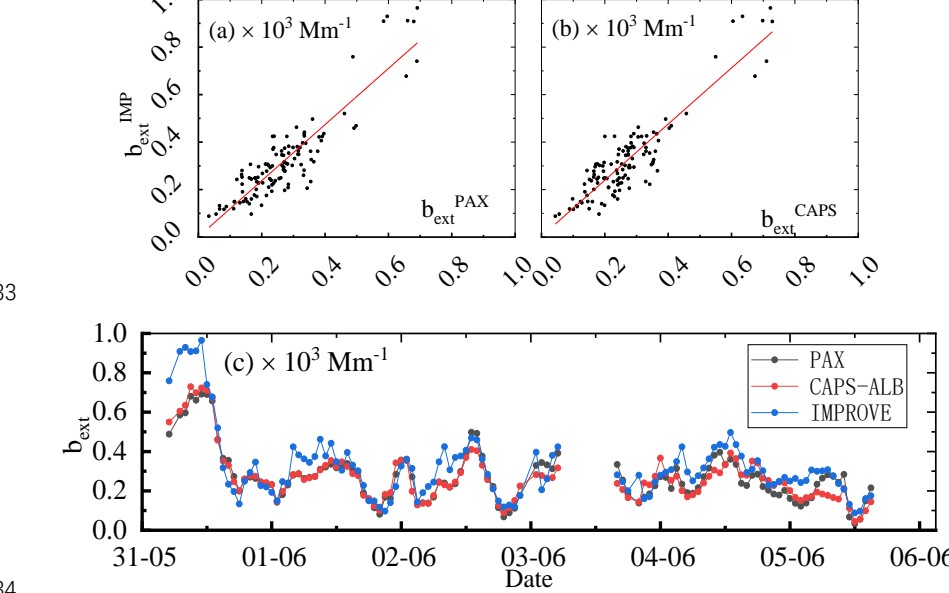



**Figure 4: Intercomparison of the measured extinction coefficient of (a) PAX and (b) CAPS-ALB with**
**IMPROVE-calculated extinction coefficient during field measurement (1-6 June 2019), and (c) the timing**
**diagram of the extinction coefficient from PAX, CAPS-ALB measurement and IMPROVE model**
**calculation.**

Then the CAPS-ALB and PAX corresponding optical properties of field measurement were
compared respectively in the case of calibrated and uncalibrated as Fig. 5(a), (b) and (c) showed. Here,
the extinction coefficient of PAX has been mentioned above as well as the absorption of CAPS-ALB was
the difference between the measured extinction and scattering. The linear fitting slope was 1.052, 1.024



and 1.046 from comparison of PAX and CAPS calibrated extinction, scattering and absorption coefficient,
with the correlation factor $R^2$ as 0.936, 0.924 and 0.772. Comparing the calibrated and uncalibrated
results, only slight corrections existed in the extinction and scattering coefficients, while the discrepancy
in the absorption coefficient has been corrected from ~30 % to less than 5 %. It can be considered that
the optical properties measured from PAX and CAPS-ALB with different measurement principles had a
good agreement, which in turn proved the reliability of our laboratory calibration results and the closure
correlation of CAPS-ALB and PAX measurements.

350       In addition, through deleting the time points of instruments data under zero calibration and abnormal

working conditions, the overall trend of calibrated extinction, scattering and absorption coefficients
during the measurement period (from 25 May to 27 June) were obtained as shown in Fig. 5(d), (e) and
(f). For the aerosol optical properties of the measurement region, it showed a dominated contribution of
scattering effect to the extinction coefficient, and a low levels of absorption coefficient.

355       The different internal structure of the nephelometers, even using the same principle, caused the

slight difference in the measured scattering coefficient. While the relatively small amount of absorption
coefficient of CAPS-ALB derived from extinction subtracted scattering coefficient has been greatly
affected. Therefore, the absorption coefficient which is difficult to quantify, was verified by CAPS-ALB
via correcting the scattering coefficient and the relationship of optical properties.



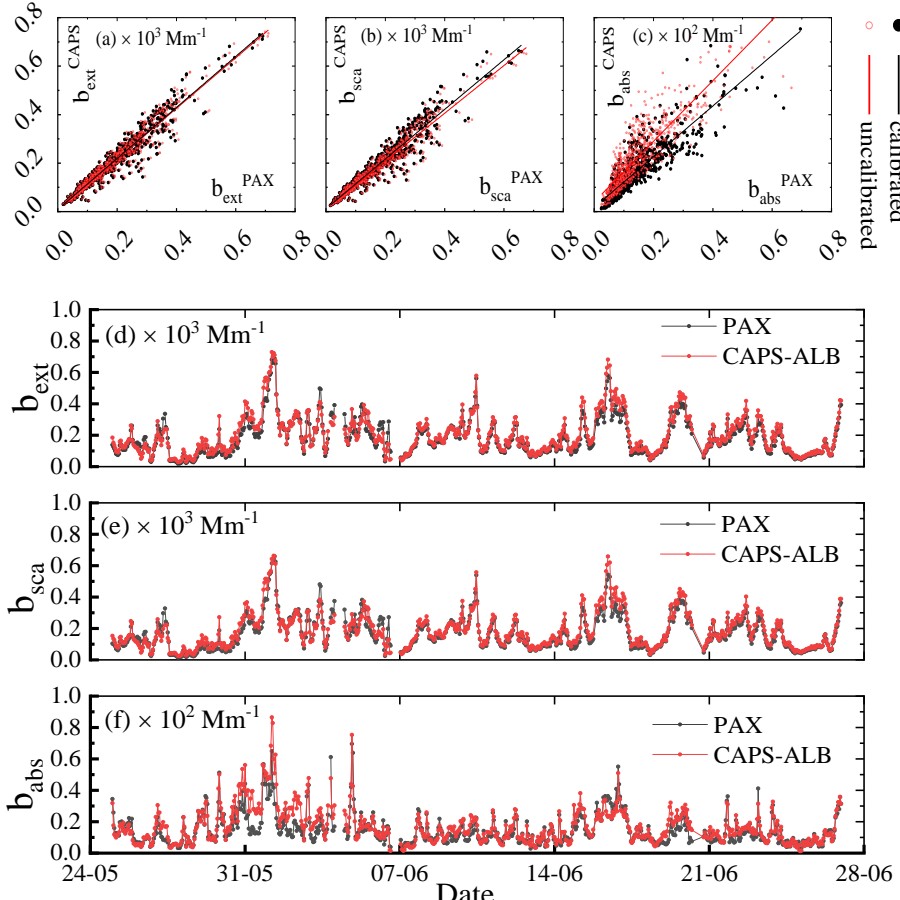

**Figure 5: Intercomparison of the CAPS-ALB and PAX (in the case of calibrated and uncalibrated) for (a) extinction, (b) scattering and (c) absorption coefficients during field measurement (From 25 May to 27 June), and the timing diagram of the calibrated (d)extinction, (e)scattering and (f)absorption coefficients of CAPS-ALB and PAX.**

## 4. Conclusion

In this work we carried out aerosol optical properties inter-comparison measurements from photoacoustic and cavity attenuated phase shift instruments. The instruments were calibrated via laboratory experiments and the corrected field measurement data have also been intercompared. Thus, following points can be concluded:

(1) The laboratory results showed that disagreements exist between the two instruments before calibration. The scattering coefficient part plays a crucial role as the bridge in constructing correlation of

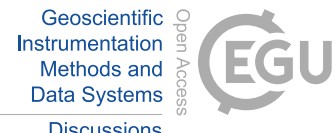

both instruments. Then the corrected extinction and absorption coefficients from both instruments were
intercompared well.
(2) The intercomparison of calibrated absorption and extinction coefficients in a field measurement
using photoacoustic and cavity attenuated phase shift instruments showed good agreement. Therefore,
laboratory calibrations were used for corrections for ensuring the quality of field data and further analysis
of radiative study.
**Data availability**
The raw data from the experiments are available upon request (j.chen@usst.edu.cn).
**Author Contribution**
**Jialuo Zhang**: Data curation, Methodology, Formal analysis, Visualization, Writing- Original draft
preparation. **Jun Chen**: Conceptualization, Investigation, Methodology, Supervision, Funding
acquisition, Writing-Review and Editing. **Meng Wang**: Validation. **Mingxu Su**: Supervision. **Wu Zhou**:
supervision. **Ravi Varma**: Investigation, Writing- Review and Editing, Methodology. **Shengrong Lou**:
Writing - Review and Editing, Validation, Funding acquisition.
**Competing Interests**
The authors declare that they have no conflict of interest.
**Financial Support**
This work was supported by the National Natural Science Foundation of China (Grant Nos. 91544225,
51776129), the National Key Research and Development Program of China (Grant Nos.
2018YFC0213800, 2017YFC0211500) and the Open Fund Project of State Key Laboratory of Loess and
Quaternary Geology (SKLLQG).

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
