# Peer review of "Intercomparison of Photoacoustic and Cavity Attenuated Phase Shift Instruments: Laboratory Calibration and Field Measurements"

_Geoscientific Instrumentation, Methods and Data Systems, 2021_

## Author Response (AR1)

**Reply to reviewer 1's comments**

Dear reviewer and editor,

Many thanks for your time to review this article. After serious consideration of your comments and suggestions, the corresponding content has been modified and supplemented. On behalf of all authors of this article, I reply to the reviewer's comments are as follows:

1. Line42: Need to describe the relation of optical properties, as "Extinction includes scattering and absorption".

The comment of reviewer have been carefully considered and the related descriptions have been added to the text.

2. Line66: What is shielding effects? How many correction factors we need? Describe the factors. Weather the "multiple scattering and shielding effects" happened in CRDS or CAPS?

The shielding effect is also called filter-loading effect, which means that as the load on the filter accumulates, the mutual shielding of the particles prevents part of the particles from being irradiated, resulting in a decrease in the measured light attenuation. The shielding effect was usually corrected by using the nonlinear relationship formula between the filter load and the light attenuation change(Weingartner et al., 2003;Arnott et al., 2005;Schmid et al., 2006;Virkkula et al., 2007;Collaud Coen et al., 2010).The multiple scattering and shielding effects are only happen in the filter-based methods, CRDS and CAPS are optical cavity spectroscopy methods, so such influence does not exist.

3. Line79-84: The description is confusing. You use particles to calibrate extinction and scattering. What is the difference?

The particles used in this study are purely scattering particles with negligible absorption, that is, theoretically, their extinction coefficient and scattering coefficient

are equal. Using the above relationship, the linear relationship between extinction coefficient and scattering coefficient can be established for reasonable correction.

4. Line99: Is IBBCEAS used to measure NO₂ concentration? Not extinction? (Line 84: "(IBBCEAS) setup was used to measure extinction coefficient of NO₂", and Line 281-282: measured extinction coefficient of ----IBBCEAS).

As shown in the following formula, IBBCEAS can indirectly measure NO2 concentration. The relationship between the $NO_2$ concentration and the extinction coefficient of each wavelength was established through the $NO_2$ extinction cross-section, and which allows the wavelength conversion of the extinction coefficient.

$$b_{ext-CEAS}(\lambda) = \left(\frac{I_0(\lambda)}{I(\lambda)} - 1\right)\left(\frac{1 - R(\lambda)}{L}\right) = \Sigma\sigma_i(\lambda)N_i$$

5. Line106: the heat was transferred to the receiving end of the instrument or the wave?

The light-absorbing components were heated and quickly transfer the heat to the surrounding air, which generate pressure wave and be detected.

6. Line101: What is the time resolution of IBBCEAS? What's the limit of detection and uncertainty in this time resolution?

As modified in the article, the time resolution of IBBCEAS was 1 min. For IBBCEAS, the limit of deteciton in this resolution was 2.4 $Mm^{-1}$ and the uncertainty was 16% mainly from the mirror -reflectivity measurement error.

7. Line282: NO₂ should be NO₂. The wavelength of CAPS-ALB was 530 nm, the wavelength of IBBCEAS was 355-380 nm, the cross-section of NO₂ was different in different wavelength range, which wavelength you used in comparison?

The comment of reviewer have been carefully considered and the modified was completed in the corresponding part of the article. The relationship between the $NO_2$ concentration and the extinction coefficient of each wavelength was established through the $NO_2$ extinction cross-section, and which allows the wavelength conversion of the extinction coefficient for the comparison with the extinciton

coefficient of CAPS-ALB at the wavelength of 530 nm.

**References**

Arnott, W. P., Hamasha, K., Moosmüller, H., Sheridan, P. J., and Ogren, J. A.: Towards Aerosol Light-Absorption Measurements with a 7-Wavelength Aethalometer: Evaluation with a Photoacoustic Instrument and 3-Wavelength Nephelometer, Aerosol Sci. Technol., 39, 17-29, 10.1080/027868290901972, 2005.

Collaud Coen, M., Weingartner, E., Apituley, A., Ceburnis, D., Fierz-Schmidhauser, R., Flentje, H., Henzing, J. S., Jennings, S. G., Moerman, M., Petzold, A., Schmid, O., and Baltensperger, U.: Minimizing light absorption measurement artifacts of the Aethalometer: evaluation of five correction algorithms, Atmos. Meas. Tech., 3, 457, 10.5194/amt-3-457-2010, 2010.

Schmid, O., Artaxo, P., Arnott, W. P., Chand, D., Gatti, L. V., Frank, G. P., Hoffer, A., Schnaiter, M., and Andreae, M. O.: Spectral light absorption by ambient aerosols influenced by biomass burning in the Amazon Basin. I: Comparison and field calibration of absorption measurement techniques, Atmos. Chem. Phys., 6, 3443-3462, 10.5194/acp-6-3443-2006, 2006.

Virkkula, A., Makela, T., Hillamo, R., Yli-Tuomi, T., Hirsikko, A., Hameri, K., and Koponen, I. K.: A simple procedure for correcting loading effects of aethalometer data, J Air Waste Manag Assoc, 57, 1214-1222, 10.3155/1047-3289.57.10.1214, 2007.

Weingartner, E., Saathoff, H., Schnaiter, M., Streit, N., Bitnar, B., and Baltensperger, U.: Absorption of light by soot particles: determination of the absorption coefficient by means of aethalometers, J. Aerosol Sci, 34, 1445-1463, 10.1016/s0021-8502(03)00359-8, 2003.

**Reply to reviewer 2's comments**

Dear reviewer and editor,

Many thanks for your time to review this article. After serious consideration of your comments and suggestions, the corresponding content has been modified and supplemented. On behalf of all authors of this article, I reply to the reviewer's comments are as follows:

1. The calibration of the instruments in the lab has both offset and multiplication factor toaccount for the drift. This means that there is an inherent absorption/ scattering evenin the absence of the absorber/ scatterer. Since CAPS and PAX are commercial instruments, such huge drifts are not expected. Can you explain if there any specific reason for the drift in the instrument calibration from the original factory specified ones?

The two reasons of drift in instrument calibration are as follows:1. due to the long-term operation of the instrument, its scattering background has deviated from the set value; 2. high concentration of absorbing gas and scattering particles was used to calibrate the instrument in this study, resulting in a correspondingly higher drift (~10%).

2. CAPS-ALB and PAX, each is running at a single wavelength (530 nm/ 532 nm). One is using an LED and the other is using a laser. Another setup, IBBCEAS instrument uses a broadband source with a CCD array spectrometer. So, in the analysis of each instrument, corresponding spectral resolution must be taken into account, especially when using gas calibration with $NO_2$ etc. What is the strategy used in this study? This must be made clear and added to the manuscript.

The comment of reviewer have been carefully considered and the related descriptions have been added to the article. For reasonable comparison in extinction coefficient of IBBCEAS and CAPS-ALB, the spectral resolution of two instruments was need to be

synchronized. CAPS-ALB uses LED as the light source and 10-nm wide optical filter to define the measurement range, but its specific band range hasn't been found, here we presumed that to be 525-535 nm. Therefore, when calculating extinction coefficient of IBBCEAS from measured $NO_2$ concentration and its absorption cross section at the specific wavelength, the average value of the $NO_2$ absorption cross section of Voigt et al. (2002) in the range of wavelength 525nm to 535nm was applied.

3. Both laboratory calibration and field measurement campaign are done in this study. It will be beneficial to add one sentence or two in the abstract regarding the field campaign undertaken.

The comment of reviewer have been carefully considered and the related descriptions have been added to the abstract "In our recent field measurement carried out in the Gehu area of southwest Changzhou City".

4. Please explain a little more about the IMPROVE model and provide relevant references.

The comment of reviewer have been carefully considered and the related descriptions have been added to the text "For comparison, the IMPROVE model was applied to identify aerosol light extinction contribution of major chemical components during field measurement. The IMPROVE model was established by analyzing the data from the long-term monitoring of aerosol mass concentration carried out in multi-site of the Inter-agency Monitoring of PROtected Visual Environments network in the United States. The IMPROVE model reconstructs extinction coefficient using the mass concentration of aerosol chemical components and their mass extinction efficiency, which has been used worldwide for estimating the aerosol extinction coefficient (Pitchford et al., 2007;Tao et al., 2014)".

5.The manuscript in general easy to read. However, it advised to have it corrected by a native speaker for proper English grammar and usage. Suggestions to correct some obvious text errors that I noticed are listed below:

a.The sentence in line 49 – 51 or page 2 has "technique" used three times. When you

specify "spectroscopy" it is interpreted as a technique in itself. Just delete the word from the sentence.

b. Lines 83-84, page 3, "Spectroscopy (IBBCEAS) setup was used …" is used. You may use "Spectrometer (IBBCEAS) was used …" instead.

c. Line 152, page 6, "self-developed" was used. I guess the authors meant that they developed it instead of a commercial purchase. If it is so, it is better to use "developed in-house" or something similar.

d. Is it "PAX" or "PAS"? Page 9, line 217.

a-c.The opinions of the reviewer have been accepted and the corresponding sentences in the text have been revised.

d.Here "PAS" in the text refers to the photoacoustic spectrometer used by Arnott et al., not the Photoacoustic Extinctiometer (PAX).

**Reference**

Pitchford, M., Maim, W., Schichtel, B., Kumar, N., Lowenthal, D., and Hand, J.: Revised algorithm for estimating light extinction from IMPROVE particle speciation data, J. Air Waste Manag. Assoc., 57, 1326-1336, 10.3155/1047-3289.57.11.1326, 2007.

Tao, J., Zhang, L., Ho, K., Zhang, R., Lin, Z., Zhang, Z., Lin, M., Cao, J., Liu, S., and Wang, G.: Impact of PM2.5 chemical compositions on aerosol light scattering in Guangzhou — the largest megacity in South China, Atmos. Res., 135-136, 48-58, 10.1016/j.atmosres.2013.08.015, 2014.

Voigt, S., Orphal, J., and Burrows, J. P.: The temperature and pressure dependence of the absorption cross-sections of NO2 in the 250-800 nm region measured by Fourier-transform spectroscopy, J. Phototech. Photobio. A, 149, 1-7, 10.1016/s1010-6030(01)00650-5, 2002.